# Understanding and maximising the community impact of seasonal malaria chemoprevention in Burkina Faso (INDIE-SMC): study protocol for a cluster randomised evaluation trial

Marta Moreno ![ORCID],[1] Aissata Barry,[2] Markus Gmeiner,[3] Jean Baptist Yaro,[2] Samuel S Sermé,[2] Isabel Byrne,[1] Jordache Ramjith,[3] Alphonse Ouedraogo,[2] Issiaka Soulama,[2] Lynn Grignard,[1] Seyi Soremekun,[1] Simon Koele,[3] Rob ter Heine,[4] Amidou Z Ouedraogo,[2] Jean Sawadogo,[2] Edith Sanogo,[2] Issa N Ouedraogo,[2] Denise Hien,[2] Sodiomon Bienvenu Sirima,[2] John Bradley,[1] Teun Bousema,[3] Chris Drakeley,[1] Alfred B Tiono[2]

MM, AB and MG contributed equally.
CD and ABT contributed equally.

For numbered affiliations see end of article.

**Correspondence to**
Dr Alfred B Tiono;
a.tiono@gras.bf

## ABSTRACT

**Introduction** Seasonal malaria chemoprevention (SMC) involves repeated administrations of sulfadoxine-pyrimethamine plus amodiaquine to children below the age of 5 years during the peak transmission season in areas of seasonal malaria transmission. While highly impactful in reducing *Plasmodium falciparum* malaria burden in controlled research settings, the impact of SMC on infection prevalence is moderate in real-life settings. It remains unclear what drives this efficacy decay. Recently, the WHO widened the scope for SMC to target all vulnerable populations. The Ministry of Health (MoH) in Burkina Faso is considering extending SMC to children below 10 years old. We aim to assess the impact of SMC on clinical incidence and parasite prevalence and quantify the human infectious reservoir for malaria in this population.

**Methods and analysis** We will perform a cluster randomised trial in Saponé Health District, Burkina Faso, with three study arms comprising 62 clusters of three compounds: arm 1 (control): SMC in under 5-year-old children, implemented by the MoH without directly observed treatment (DOT) for the full course of SMC; arm 2 (intervention): SMC in under 5-year-old children, with DOT for the full course of SMC; arm 3 (intervention): SMC in under 10-year-old children, with DOT for the full course of SMC. The primary endpoint is parasite prevalence at the end of the malaria transmission season. Secondary endpoints include the impact of SMC on clinical incidence. Factors affecting SMC uptake, treatment adherence, drug concentrations, parasite resistance markers and transmission of parasites will be determined.

**Ethics and dissemination** The London School of Hygiene & Tropical Medicine's Ethics Committee (29193) and the Burkina Faso National Medical Ethics Committee (Deliberation No 2023-05-104) approved this study. The findings will be presented to the community; disease occurrence data and study outcomes will also be shared with the Burkina Faso MoH. Findings will be published irrespective of their results.

**Trial registration number** NCT05878366.

## STRENGTHS AND LIMITATIONS OF THIS STUDY

⇒ Strong engagement with National Malaria Control Program in Burkina Faso to tailor data collection to local needs.
⇒ This study quantifies for the first time the effect of seasonal malaria chemoprevention (SMC) on the human infectious reservoir for malaria.
⇒ The trial uses measurements of drug plasma concentrations as direct measurements of the effective doses of SMC drugs that are achieved.
⇒ Potential impact on SMC uptake due to presence of study teams is mitigated by a cross-sectional study survey outside of the study area.
⇒ Study conduct is limited to a single study site.

## INTRODUCTION
### Background

Efforts to reduce the global incidence of malaria are failing; there were an estimated 247 million cases of malaria worldwide in 2021, up from 230 million in 2015.[1] With the threat that control efforts are further compromised by insecticide resistance and parasite resistance to first-line drugs, there is a clear need for interventions that reduce the clinical burden of malaria and onward parasite transmission.[2]

One recent intervention that has the potential to profoundly reduce the incidence of (severe) clinical disease is seasonal malaria chemoprevention (SMC).[3] SMC provides antimalarials to the most vulnerable populations during the peak of malaria transmission season with the aim to reduce disease burden in this high-risk population. To date,

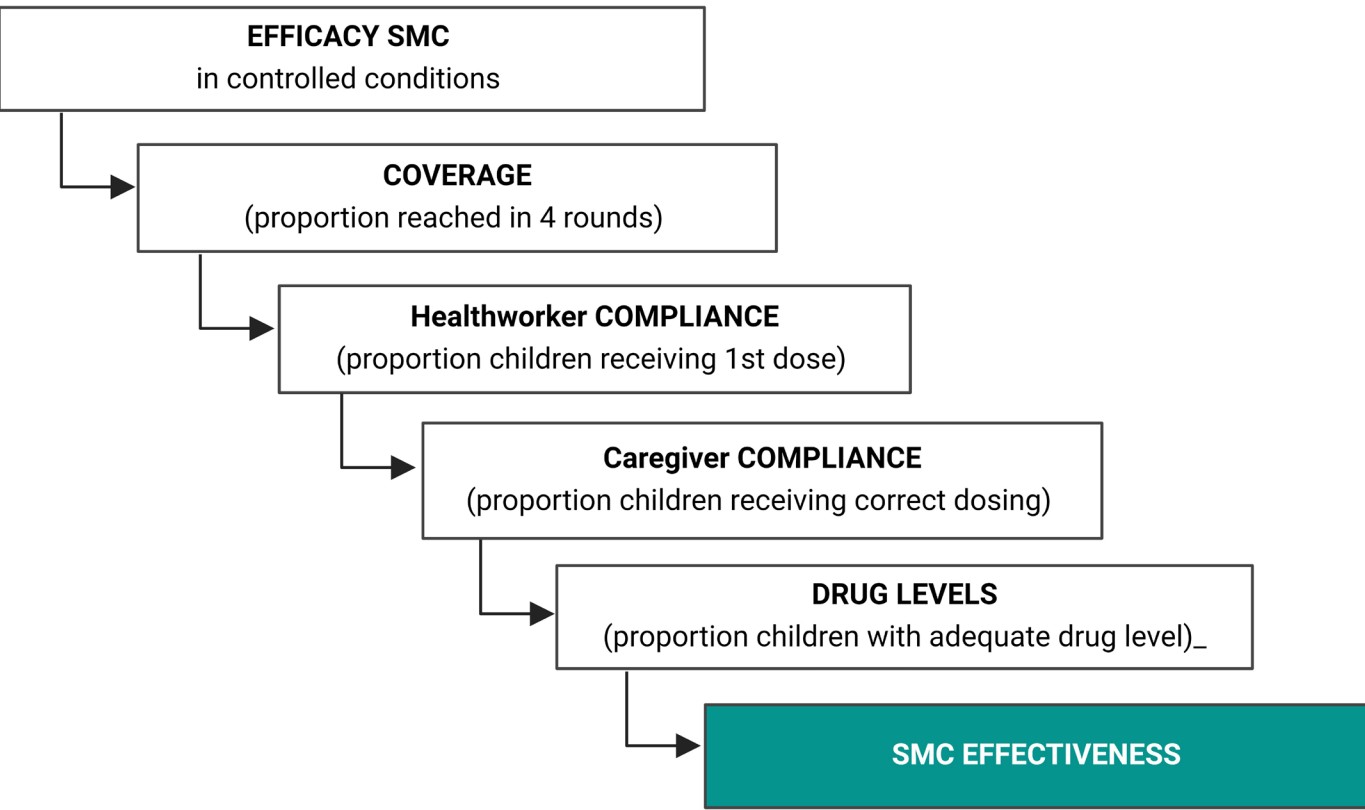

**Figure 1** Cascade in efficacy decay in seasonal malaria chemoprevention (SMC) effectiveness. Coverage, compliance and drug absorption will all influence the actual efficiency that is observed when SMC is implemented under programmatic conditions.[28] DOT, directly observed treatment.

SMC has been administered to millions of African children.[4 5] Although most commonly given to children below the age of 5 years, SMC is recommended for all children in age groups that are at high risk of severe malaria.[6] The recent WHO guidelines of the year 2022 offer increased flexibility in defining the specific number of monthly cycles and the target age group. In Burkina Faso, SMC is currently administered over four rounds to all children aged 3–59 months in July, August, September and October. These SMC rounds cover the peak of the malaria transmission season that follows seasonal rains that typically peak in July to September. Several countries in the Sahel region, including Burkina Faso, are considering extending the target age range of SMC to all children under the age of 10 years.

While SMC is highly efficacious in research settings, reducing morbidity by up to 60%,[5 7] effect sizes may be lower when SMC is rolled out programmatically.[5 8 9] A cascade in efficacy decay is regularly observed for efficacious interventions; for SMC this efficacy decay may be related to coverage, compliance (at multiple levels), drug absorption and parasite resistance to SMC drugs (figure 1). A large study that assessed plasma levels of SMC drugs in children in Niger reported evidence for incomplete adherence or drug absorption in >80% of children.[9] The choice of drug regimen and parasite resistance to drug components may also play important roles. SMC with the conventional drug combination of

sulfadoxine-pyrimethamine plus amodiaquine (SPAQ) is not always effective at clearing parasites: up to 46% of children under the age of 5 years may carry parasites at the end of the transmission season.[10 11] Recently, longitudinal studies in Burkina Faso and The Gambia observed that a large proportion of SMC-targeted children either remained *Plasmodium falciparum* infected or became reinfected shortly after receiving SMC with SPAQ.[11 12]

Parasite resistance to the SPAQ antimalarial combination may play a role in defining its anti-infection effects. Resistance to SP is strongly associated with the presence and number of point mutations in dihydropteroate synthase (dhps; eg, mutations 437Gly and 540Glu) and dihydrofolate reductase (dhfr; eg, 51Ile, 59Arg and 108Asn) genes.[13 14] The prevalence of these core mutations and the number of additional mutations in dhps and dhfr are rising across Africa. In East Africa, quintuple mutations in dhfr-dhps are common although a single study suggests that SMC with SPAQ retains protective efficacy.[15] Compared with resistance markers for SP, the molecular mechanisms underlying AQ resistance are less well understood and molecular markers for AQ have limited predictive value.

AQ is metabolically activated, primarily by Cytochrome P450 isoenzyme 2C8 (CYP2C8), into desethylamodiaquine (DEAQ). The $T_{max}$ for AQ is around 2 days (IQR 1–3 days) and drug concentrations wane rapidly between days 3 and 7. DEAQ concentrations peak ~2 days after

AQ and the terminal half life of DEAQ (~196.4 hours) is approximately fourfold longer than that for AQ.[16] AQ resistance associates with variant haplotypes in chloroquine transporter (crt; eg, 72Cys-73Val-74Ile-75Glu-76Thr) and multidrug resistance gene 1 (mdr1; eg, encoding Tyr at codons 86, 184 and 1246)[17] but better markers may become available.

In addition to imperfect efficacy against (incident) asexual-stage parasites, SPAQ may also be unable to prevent parasite transmission shortly after treatment. Gametocytes typically persist for several weeks after SPAQ[18] with conflicting findings on gametocyte infectivity after SP treatment. In vitro and in vivo studies have demonstrated considerable gametocyte sterilisation by SP where gametocytes are detectable but no longer (as) infectious.[19 20] However, clinical trials in naturally infected gametocyte carriers demonstrated that SPAQ is ineffective in clearing gametocytes and individuals may remain infectious to mosquitoes for several weeks after treatment.[21 22] Adding single low-dose primaquine rapidly prevented transmission shortly after SPAQ.[21]

With SMC being deployed at community level, this intervention has the potential to reduce community transmission and thereby reduce the force of infection experienced by the general population. Recent work directly assessing the infectivity of parasite carriers to mosquitoes identified school-age children (aged 5–15 years) as important reservoirs of infection.[23–26] This population is less likely to become symptomatic on infection and supports high parasite densities and transmissible gametocytes.[23–26] Younger children are comparatively less important because their infections are more likely to receive treatment that prevents chronic parasite carriage and due to their small body size and high bed net coverage, mosquito exposure can be relatively low.[27] In contrast, school-age children are heavily exposed, currently undertargeted with conventional interventions (eg, have lowest bed net coverage) and carry the highest densities of gametocytes.[23 28] While less at risk of severe clinical disease, also school-age children may suffer from consequences of parasite carriage that are often overlooked, including haemolysis, increased susceptibility to bacterial infections[29] and reduced cognitive function[28]; also this age group would thus likely benefit from interventions that reduce this parasite carriage. Extending the target age range for SMC from all children below 5 years of age to all children below 10 years of age will thus increase the population experiencing direct personal benefits from treatment and may also potentially reduce community-wide transmission by covering a large proportion of the human infectious reservoir for malaria. Further extending the age range for SMC to include all children below 15 years of age is currently not considered by National Malaria Control Program (NMCP).

Recent observations lead to questions related to the reasons for the continued high infection rates in the SMC-targeted population and the potential impact of SMC when the age range is extended to populations that are important to fuelling transmission to the wider community.

## Rationale

It is currently unclear whether failure of SMC with SPAQ to prevent parasitaemia is associated with (1) imperfect adherence, (2) parasite resistance to the drug components, (3) active drug concentrations that are achieved in SMC participants and/or (4) consistent exposure to reinfection. It is also unclear whether extending the targeted age range will reduce the infectious reservoir for transmission to a level where community benefits may be expected. Lastly, factors that determine efficacy decay are currently poorly understood. Quantifying each of these elements should contribute to understanding the effectiveness of SMC in programmatic settings.

## Objectives

This study aims to collect detailed information on parasite carriage, gametocyte carriage and infectivity in children targeted and untargeted by SMC in Burkina Faso. The impact of expanding SMC to include children up to 10 years of age on reducing the human infectious reservoir and factors driving imperfect antiparasite or antigametocyte effects of SMC will be assessed. The infectious reservoir will be examined using a combination of gametocyte density estimates in the entire population and mosquito membrane feeding assays in a selection of participants. Our specific aims are to:

► Compare SMC effectiveness in children aged 3–59 months in preventing clinical malaria and asymptomatic parasite carriage when implemented by the NMCP or when implemented in a research context where all doses are directly observed.
► Evaluate the effectiveness of SMC to reduce malaria infection prevalence in 5–9 year-olds.
► Quantify the infectious reservoir and the contribution of different age groups to transmission with conventional SMC (<5 years) and extended SMC (<10 years).
► Determine the impact of drug resistance (by molecular typing of *dhps*, *dhfr*, *pfcrt* and *pfmdr1*) and drug absorption on SMC efficacy.
► Understand social barriers and enablers interfering with SMC efficacy and how SMC uptake is related to health equity with special attention to gender inequalities.
► Quantify SMC efficacy decay under programmatic conditions and key drivers of this decay.

## METHODS AND ANALYSES
### Trial design

The study is designed to evaluate SMC effectiveness and consists of three study arms, with the first arm receiving an intervention that is implemented as part of routine control. This study involves an operational evaluation of a modified existing intervention delivered in independent clusters. The project and its implementation are prepared in direct interaction with the Burkina Faso Ministry of

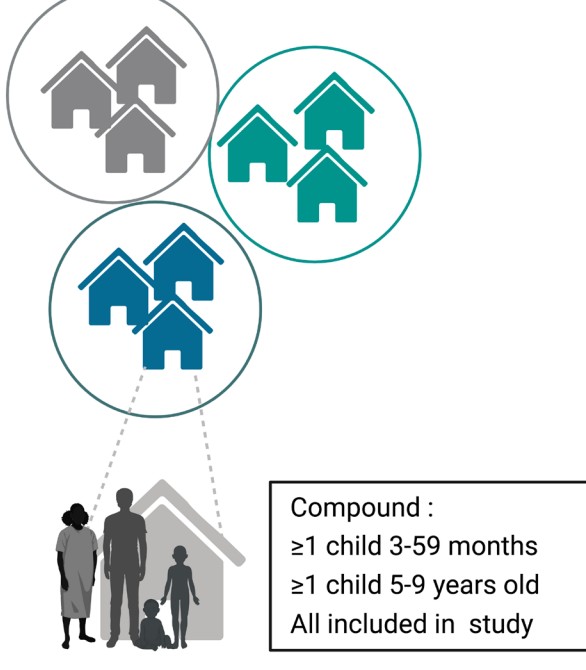

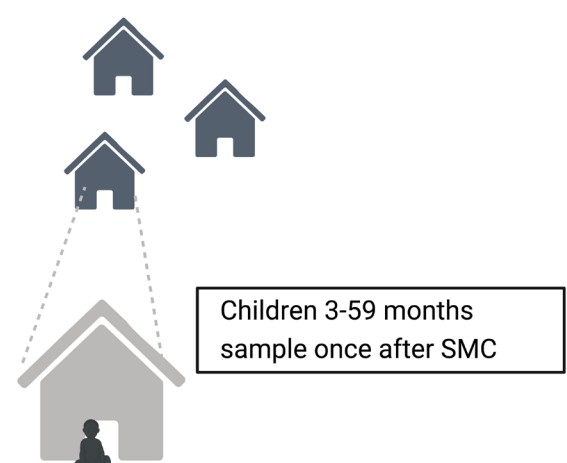

**Figure 2** Study design. The study will be implemented in clusters. Each cluster consists of three compounds where each compound has at least one child aged 3–59 months and one child aged 5–9 years. All age groups will take part in the evaluation of the study to assess the impact of the intervention on the infectious reservoir in the community and compare parasite carriage in targeted and untargeted populations. SMC, seasonal malaria chemoprevention. (Figure was created with BioRender.com.)

Health (MoH) and NMCP to tailor data collection to local needs. There will be three study arms:

*Study arm 1* (*control*): SMC in children under the age of 5 years, implemented by the MoH; the first dose is given under supervision but there is no directly observed treatment (DOT) for the full course of SMC.

*Study arm 2* (*intervention*): SMC in children under the age of 5 years, with DOT for the full course of SMC.

*Study arm 3* (*intervention*): SMC in children under the age of 10 years, with DOT for the full course of SMC.

SMC consists of SP and AQ administered daily for 3 days.[30] SP is given on the first SMC day only while AQ is given daily over 3 days. In total, four rounds of SMC are conducted during the peak of malaria transmission season from July to October, separated by approximately 30 days, concurrently with programmatic interventions in study arms 2 and 3.

Each study arm contains 61–62 clusters; each cluster consists of three compounds where each compound has at least one child aged 3–59 months and one child aged 5–9 years (185 compounds per arm; 555 compounds in total) (figure 2). The total number of enrollees per arm is the primary statistical consideration; cluster size (three compounds per cluster) was decided on based on prior experience[11] and logistical feasibility. The cluster design is taken into consideration in statistical analyses. All age groups will take part in the evaluation of the study to assess the impact of the intervention on the infectious reservoir in the community and compare parasite carriage in targeted and untargeted populations. To detect possible behavioural contamination and differences in SMC uptake or adherence in study arm 1 compared with what would be observed if there would be no study team present, a single cross-sectional survey will take place 7 days after the last (fourth) round of SMC outside the study area (2–5 km from the nearest enrolled study compound). SMC coverage is assessed by questionnaire and sample collection to determine plasma levels of SMC drugs and parasite carriage.

This protocol was developed following the Standard Protocol Items: Recommendations for Interventional Trials reporting guidelines.[31]

## Box 1 Study primary, secondary and exploratory endpoints

**Primary endpoint**
⇒ Parasite prevalence by quantitative PCR (qPCR) at the end of the transmission season in age groups targeted by seasonal malaria chemoprevention.

**Secondary endpoints**
⇒ Parasite prevalence by qPCR at the end of the transmission season in all age groups (between arm comparison).
⇒ Parasite prevalence by microscopy prior to seasonal malaria chemoprevention (SMC) rounds 2, 3 and 4 in SMC-targeted age groups (between arm comparison).
⇒ Rate of reinfection with *Plasmodium falciparum* at weeks 3, 4 and 5 after the last round of SMC, assessed in SMC-targeted age group*s* (between arm comparison).
⇒ Gametocyte prevalence by quantitative reverse transcriptase (qRT)-PCR at weeks 3, 4 and 5 after the last round of SMC, assessed in SMC-targeted age groups (between arm comparison).
⇒ Gametocyte prevalence by qRT-PCR at the end of the transmission season in all age groups (between arm comparison).
⇒ Gametocyte prevalence by qRT-PCR at the end of the transmission season in age groups targeted by SMC (comparison between arms 1 and 2 (in children aged 3–59 months) and arms 2 and 3 (in children aged 5–9 years)).
⇒ Plasma levels of amodiaquine (AQ) and desethylamodiaquine (DEAQ) after the fourth round of SMC in children aged 3 months to 9 years (between arm comparison and comparison with control area).
⇒ Incidence of clinical malaria captured during passive case detection (between arm comparison).

**Exploratory endpoints**
⇒ Infectivity to mosquitoes, defined as the percentage of infected mosquitoes, in relation to gametocyte density and plasma drug levels of AQ and DEAQ (across arms assessment).
⇒ Size and age distribution of the infectious reservoir for malaria, defined as the likelihood that a mosquito becomes infected with malaria parasites after feeding on a population member (between arm comparison).
⇒ Prevalence of drug resistance markers in infected children aged 3 months to 9 years assessed after each round of SMC (between arm comparison).
⇒ Description of perceived social barriers to SMC uptake.
⇒ Quantification of SMC efficacy decay under programmatic conditions.

## Endpoints

The primary endpoint is parasite prevalence by quantitative PCR (qPCR) at the end of the malaria transmission season in age groups targeted by SMC. This endpoint will be compared between study arms 1 and 2 (in children aged 3–59 months) and arms 2 and 3 (in children aged 5–9 years). Additional secondary and exploratory endpoints will be assessed (box 1).

## Study sites

The study site is in Saponé Health District (45 km southwest of Ouagadougou) and the study will benefit from historical records on malaria burden in this area and research infrastructure that was established in previous joint research projects.[24 32 33] Up-to-date census data will be used prior to the study to select eligible compounds for enrolment.

## Study population

The study population will be all members of 555 compounds (61–62 clusters per study arm; each cluster consisting of three compounds) in Saponé Health District, Burkina Faso. Clusters are randomised over three study arms. Compound members of all ages will be included in study procedures (eg, evaluation of the intervention) but only children will be targeted with the intervention. Based on census data, and recent studies in the same study area, we expect an average of 1.5 children aged <5 years and 1.5 children aged 5–9 years per compound (4.5 children aged <5 years and 4.5 children aged 5–9 years per cluster, respectively).

## Study duration

Total duration of the trial is 13 months, from July 2023 to July 2024.

## Inclusion and exclusion criteria

The inclusion and exclusion criteria (table 1) for the evaluation trial are broad to ensure generalisability of the collected data and results.

## Recruitment

Following community meetings where study objectives and procedures are explained, study teams visit individual households to explain study objectives and procedures for a second time—this time to the head of household and all household members of assenting or consenting age. Study information is presented orally and printouts in French and/or a local language are provided. The voluntary nature of study participation and the right to withdraw at any time are emphasised. It is explained that refusal to participate will not affect access to SMC or any other medical treatment. All household members are encouraged to ask questions prior to signing informed consent (aged >19 years) or assent (aged 12–19 years). Forms and participant information sheets are included as online supplemental material 1.

## Patient and public involvement

One month prior to the study roll-out community meetings were organised to discuss study objectives and procedures. Special emphasis was given to participant-facing activities, including informed consent. Study findings are presented in community meetings and discussed in depth with community-based opinion leaders.

## Sample size
### Clusters

Evaluable study clusters will comprise three compounds and have between four and five children below 5 years of age (eligible for conventional SMC in all study arms) and similar numbers of children aged 5–9 years (eligible for extended SMC in arm 3). We assume ~40% parasite prevalence detected by qPCR at the end of the transmission

**Table 1** Inclusion and exclusion criteria for the evaluation trial and for the qualitative study on SMC acceptability and uptake

| | |
|---|---|
| Compounds are eligible for participation in the trial if: | there are at least 1 child aged <5 years and 1 child aged 5–9 years who are permanent residents of the compound and willing to participate. |
| | residents are willing to provide repeated blood samples up to a total volume of 16 mL during the entire study period (maximum of 4 mL for children aged <5 years). |
| | the eligible individual or caregiver(s) provide informed consent and additional assent is provided from minor participants aged 12–19 years. |
| Household members are excluded from participation in the trial in case of: | any (chronic) illness that requires continuous care. |
| | current participation in malaria vaccine or drug trials or participation in vaccine trials in the last 2 years. |
| | ineligibility for SPAQ administration according to NMCP guidelines (only for children). |
| Community members are eligible for participation in the qualitative study if they are: | a resident of a household participating in the trial. |
| | caregiver of a child eligible to SMC. |
| | 20 years of age or older. |
| Healthcare workers are eligible for participation in the qualitative study if they are: | serving as a healthcare worker delivering SMC intervention. |
| | serving as health staff in the study zone. |
| Community members are excluded from participation in the qualitative study if they are: | not residing in the household during SMC preceding the qualitative survey. |

NMCP, National Malaria Control Program; SMC, seasonal malaria chemoprevention; SPAQ, sulfadoxine-pyrimethamine plus amodiaquine.

season in children aged 5–9 years without SMC and 16% parasite prevalence in children aged <5 years under programmatic SMC (Collins *et al*).[11] When assuming a reduction to 25% parasite prevalence in children aged 5–9 years when included in SMC, a sample size of 55 clusters per arm (×3 compounds per cluster and an estimated total six children in this age group per cluster) will achieve 87% power to detect this difference at an alpha of 0.05 and a coefficient of variation of 0.3 (Collins *et al*).[11] This sample size will give over 80% power to detect a reduction of parasite prevalence in directly observed SMC to 7% as compared with 16% under programmatic SMC, in children aged <5 years. The total number of compounds per arm has been increased to 62 to mitigate the effects of non-compliance and drop-outs and for logistical reasons, such entire villages are included. We will thus enrol a total of 185 clusters over three arms in a ratio of 1:1:1 with 555 compounds in total. We anticipate that this will include a minimum of 720 children aged <5 years and 720 children aged 5–9 years (based on recent census data). The cluster design of the intervention and evaluation will be accounted for in the statistical analysis.

## Human infectious reservoir assessments

For assessments of the human infectious reservoir, we use a combination of direct measurements of transmission by mosquito feeding assays and imputation of transmissibility based on gametocyte density.[25] The reason for a partial reliance on imputation is that the intervention study is too large in scope and the time window for mosquito feeding experiments is too narrow (ie, mosquito feeding is ideally done within weeks after SMC) to assess a direct impact of SMC on gametocyte infectivity with a conventionally powered comparison between arms. We will thus use a combination of methodologies where we quantify the association between gametocyte density and infectivity in the absence of treatment, demonstrating consistency across settings,[23 24 34] and impute mosquito transmission from gametocyte density.[23 25] Direct comparisons of the association between gametocyte density and mosquito infection rates in mosquito feeding assays can demonstrate whether imputation is indeed acceptable across populations[23] or whether there are subpopulations with a different association (eg, due to gametocyte-sterilising effects of antimalarials).[35] Based on these insights, direct mosquito feeding assays (DMFAs) will be performed on gametocyte-positive individuals with and without prior SMC and mosquito infection rates for a given gametocyte density will be compared.[35] Since we hypothesise that SPAQ may permanently sterilise gametocytes, any reduced infectivity of gametocytes will remain apparent even after drug levels have waned (ie, the gametocytes are permanently damaged by drug exposure).[19] We can efficiently assess this possible sterilising effect of SPAQ by preferentially recruiting children who are gametocyte positive at the start of SMC and invite them for mosquito feeding after SMC. This will be done in children aged 5–9 years only who carry the highest parasite and gametocyte densities; this also avoids phlebotomy from the youngest children. We anticipate a median gametocyte density of ~14 gametocytes/µL (IQR 1.8–44.2) as observed post-treatment.[20 36] While the number of feeding assays will be informed by feasibility (eg, mosquito husbandry), we provide an indication of study power. When recruiting 100 children who received SPAQ (arm 3) and 100 children

who did not (arms 1 and 2) and dissecting 40 mosquitoes per experiment, simulations show we would have >90% power to detect a 70% reduction in infectivity.[37]

## Outside study area survey

To give a broad indication of the potential influence the presence of the study team may have on the operational delivery of SMC in arm 1, we will perform a single cross-sectional survey outside the study area. The survey is not part of any main comparisons, and an exact sample size justification is complicated. Estimates of adherence are based on questionnaire data and supported by measurements of drug levels in plasma. In arms 2 and 3 (DOT for the full course of SMC), the median level of DEAQ at day 4 following the last dose of AQ is expected to be 550 ng/mL (IQR 350–850).[16] Based on a large assessment of reported adherence and measured AQ/DEAQ plasma levels following SMC in Niger, we anticipate <50% of children show complete adherence in arm 1 and the control village and >20% of the children in these populations may fail to take any dose.[9] If we conservatively assume that this results in a reduction of median DEAQ levels to <400 ng/mL (IQR 200–700), we can estimate the power to detect this difference when enrolling 120 children in the SMC-targeted age range in the control area. For this, we assume that these DEAQ levels stem from right-skewed (log-normal) distributions; giving median log DEAQ concentrations of 6.31 (IQR 5.86–6.75) for DOT and 5.99 (IQR 5.30–6.55) for non-DOT. If we further assume that the distribution of the logged values is normally distributed, we can estimate the SD through the IQR as IQR/1.349, giving SDs of 0.93 and 0.66, respectively. Using a sample size of 120 per group, and a Welch's two-sample t-test for testing the differences between the log DEAQ concentrations, we estimate a power of 86.3% to

detect these differences. Based on these assumptions, we are confident that sampling 120 children <5 years of age will allow meaningful assessments. As indicated above, this sample size estimation is not relevant for any of the main comparisons in the study but merely used to ensure informative sampling and avoid exposing an unnecessarily large population to the small discomfort of a single fingerprick sample.

## Study procedures

### Seasonal malaria chemoprevention

SMC drug administration uses SPAQ at the conventional dosing (online supplemental material 2). This medication is provided by the MoH. SMC is administered by MoH teams in arm 1 (no DOT) and by dedicated research staff in arms 2 and 3 (DOT). Clinical malaria cases will receive conventional first-line treatment with artemether-lumefantrine (AL) according to national treatment guidelines when they attend study clinics.

### Scheduled surveys

Scheduled surveys will take place in each study arm and are designed and timed for several outcome measures (figure 3):
1. A survey prior to SMC roll-out (baseline survey) is used to determine parasite carriage and to collect parasite material for genotyping infections and comparing parasite genotypes with parasites detected following SMC.
2. A mid-study survey aims to assess parasite carriage at different time points following SMC and relate this to time since SMC and SMC plasma levels. These surveys are scheduled 3, 4 or 5 weeks after the last round of SMC. Each compound will participate in one of these surveys; each cluster has three compounds and one of these compounds is randomly selected to par-

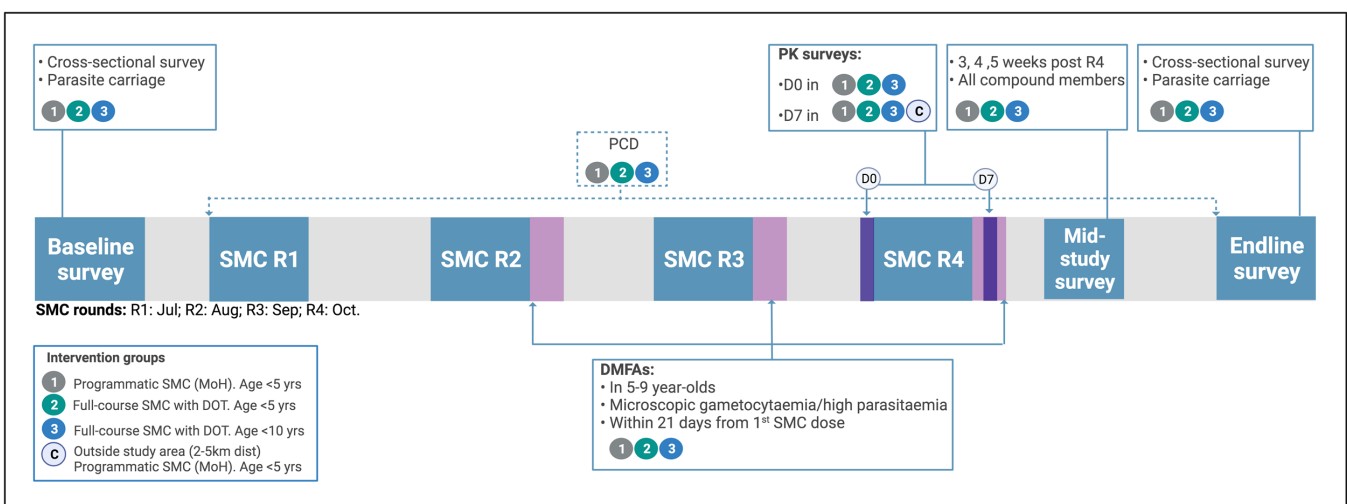

**Figure 3** Surveys and sampling schedule. Cluster randomisation was performed prior to the baseline study. Adherence to and perceptions about seasonal malaria chemoprevention (SMC) strategy will be collected throughout the study in the three study arms. DMFA, direct membrane feeding assay, done up to 21 days from first SMC dose; DOT, directly observed treatment; MoH, Ministry of Health; PCD, passive case detection; PK surveys, pharmacokinetic analysis. (Figure was created with BioRender. com.)

ticipate in the first, second or third mid-study survey. These surveys will include all compound members (all age groups) to also allow comparisons of gametocyte carriage and transmission potential between arms.

3. The end of study survey (endline survey) will target all age groups in the study households to determine whether parasite carriage and gametocyte carriage are reduced both in the SMC-targeted populations and in the wider community. This survey will also provide information on the primary endpoint (parasite carriage by qPCR at the end of the transmission season). During these surveys, all individuals will have their axillary temperature measured and a fingerprick blood sample taken onto a dried blood spot and into an EDTA-coated microtainer for collection of plasma and nucleic acids for later RNA work and DNA work. In case of measured fever, the same fingerprick blood sample will also be used for malaria diagnosis by conventional rapid diagnostic test to guide treatment according to national guidelines.

4. In addition to these surveys, there will be assessments of drug metabolites (shortly after the last round of SMC). Screening for parasite carriage by microscopy in children aged 5–9 years is scheduled on the first day of SMC rounds 2–4. Microscopy-positive gametocyte carriers will be prioritised for mosquito feeding assays (DMFA) to assess infectivity. Other microscopy-positive non-gametocyte carriers may be included for DMFA if mosquito numbers allow. DMFA will occur within 21 days after the first dose of SMC.

### Passive case detection

In addition to these scheduled surveys and throughout the study period there is malaria passive case detection in all four health facilities that are accessed by the selected study population. Participants under 10 years will receive ID cards to facilitate identification. Trained clinical personnel are available 24/7. In case of clinical malaria, defined as microscopy-confirmed malaria infection with measured or reported fever, treatment is provided with AL, according to MoH policy. Before treatment, a fingerprick blood sample is taken for parasitological and drug metabolite assessment. Clinical incidence will be expressed per person-month in the target population (<5 years old in arms 1 and 2 and <10 years old in arms 2 and 3).

### Adherence to and perceptions about SMC strategy

Selected community members and health workers will be invited for participation in the qualitative part of the study. The qualitative component of this study is designed to understand potential factors that influence SMC uptake and effectiveness, including:

1. SMC coverage—defined as proportion of eligible children reached by the four rounds of SMC.
2. Health workers' compliance to MoH guidelines (proportion of children receiving a DOT for the first dose

of SMC drugs as recommended by the treatment guidelines).

3. Caregivers' compliance—defined as the proportion of children receiving the correct dosing regimen (number of tablets and duration).
4. Perceptions of the SMC programme from caregiver and health system perspectives.

Perceptions will include the perceived benefits and ease of participation of SMC, as well as perceived utility of additional interventions (eg, long-lasting nets, indoor residual spraying, personal protective measures) when adhering to SMC.

For participation in the qualitative study on SMC adherence and perceptions, there are distinct inclusion and exclusion criteria for community members and health workers (table 1).

Throughout the study, data on compliance to SMC treatment guidelines from caregiver and healthcare worker perspectives are collected in arm 1. Whether first dose was directly observed and whether the full treatment course was administered to the child after each round on SMC is documented. After SMC round 4 a cross-sectional survey will be carried out. For this, individual interviews of caregivers of children targeted or not by the current SMC strategy will be conducted. Further, within each of the study arms, between six and eight individuals will be recruited for focus group discussions.

Key informant interviews (KIIs) and in-depth interviews (IDIs) will be carried out. For the KIIs, participants will be recruited through purposive sampling to include stakeholders within the health system involved in SMC implementation. The IDIs will be conducted among 5–10 key informants per each of the study arms. These individuals will also be purposively selected aiming at including community leaders, professionals or residents who have first-hand knowledge about the community.

### Laboratory assessments
#### Parasitology

Thick and/or thin blood films for parasite counts will be obtained at the time point of the first SMC dose for rounds 2–4 during clinical malaria episodes and during cross-sectional sampling to select study participants for assessment of infectivity by DMFA.

Further, parasite DNA and RNA will be extracted from whole blood samples in EDTA tubes and tested using qPCR that targets varATS and detects all *P. falciparum* parasites (asexual parasites and gametocytes). Parasite prevalence and density will be assessed as intervention outcome measures, as previously done for community interventions.[32]

*P. falciparum* gametocytes will be quantified by quantitative reverse transcriptase PCR targeting female (Ccp4) and male gametocytes (PfMGET).[38 39] For a subset of infections, multiplicity of infection and the genetic relatedness of infections that are detected in the same study participant over time will be determined (eg, AMA-1 amplicon deep sequencing). This amplicon sequencing

will be performed in a selection of DNA samples that are parasite positive by varATS qPCR and for whom reinfection or infection persistence is suspected.

Assessments of molecular markers of resistance will follow established protocols.[14] Nucleic acids will be extracted with commercial kits (Qiagen) and stored at −20°C until use. Three dual-labelled probes were designed to detect three crt genotypes at codons 72–76 (encoding Cys-Val-Met-Asn-Lys (CVMNK), CVIET and SVMNT). Laboratory isolates *P. falciparum* 3D7, Dd2 and 7G8 will be used as positive controls for the CVMNK, CVIET and SVMNT haplotypes, respectively. qPCR amplification will be done in a single well to maximise throughput using a thermocycler (Bio-Rad). All samples will further be tested for mdr1, dhfr and dhps. For this, all three genes will be amplified by nested PCR using previously described methods (Pearce *et al*, 2003)[40] and polymorphisms identified by direct sequencing of amplified products (BigDye Terminator v3.1 cycle sequencing kits and ABI 3730 sequencer; Thermo Fisher Scientific), and data will be analysed using Geneious V.10.1.3 (Biomatters, San Diego, California, USA).

### Pharmacology
AQ and DEAQ concentrations will be quantified from plasma samples and filter paper spots. Blood concentrations are measured within 6 months of sample collection using a liquid chromatography tandem mass spectrometry assay.[9 41]

### Serology
Serological analysis of these samples will be conducted using bead-based assays to assess:
► Presence of biomarkers for correlates of infection and infectivity (ie, proteins associated with inflammation) or parasite density (HRP2).[42]
► Antibody responses to gametocyte and asexual stage proteins in established multiplex platforms.[42] Antigens will include but not be restricted to Pfs230, Pfs48/45 for sexual stage proteins to assess the influence on transmission to mosquitoes and PfAMA, PfMSP-1 and PfETRAMP-5 to assess the effect of interventions on the development of humoral immunity.

### Mosquito feeding assays
For each assessment of infectivity, 2 mL of heparinised venous blood will be drawn from the study participant selected for DMFA. At the insectary, ~70 *Anopheles gambiae sensu stricto* mosquitoes will be fed on the subjects' blood for 15–20 min using a water-jacketed feeder system.[32] Mosquitoes will be dissected in 0.5% mercurochrome on the seventh day after the feeding assay for the prevalence of mosquitoes with oocysts and quantification of oocysts.

### Randomisation
The unit of randomisation is the cluster; each cluster comprises three compounds. To account for spatial differences in malaria exposure, distance from health facilities and treatment-seeking behaviour, neighbouring clusters are grouped before randomisation. Clusters are randomly allocated to one of the three study arms by a computer-generated algorithm and stratified by village. In this way, clusters are spatially 'matched' and are thus likely to experience similar malaria exposure. In total, the randomisation resulted in 185 compounds (61–62 clusters with three compounds each) in each arm.

### Analyses plan
#### Primary endpoint
The primary endpoint on parasite prevalence (by qPCR) at the end of the transmission season in age groups targeted by SMC will be compared in children aged 3–59 months in study arms 1 and 2 and in children aged 5–9 years in study arms 2 and 3. Parasite prevalence by qPCR will be used as binary variable with all parasite densities above 100 parasites/mL (0.1 parasite/µL) being classified as parasite positive. Parasite prevalence is compared between arms by logistic regression; models will account for clustering by including compound ID as a random intercept and village as a fixed effect.

#### Secondary and exploratory endpoints
A complete statistical analysis plan has been developed for this trial and is provided in online supplemental material 3. All parasite prevalence-related endpoints will be compared as described above for the primary endpoint. Gametocyte prevalence-related endpoints will be presented as prevalence estimate with 95% CI.

### ETHICS AND DISSEMINATION
The study has been reviewed and approved by the London School of Hygiene & Tropical Medicine (Review No 29193; version 2.0, July 2023) and Burkina Faso National Medical Ethics Committees (Deliberation No 2023-05-104). Data will be stored on secure servers at GRAS and shared with primary project partners as deidentified. A study participant identification link log is available at the investigators' site. Data will be deposited with publications and uploaded to shared data repositories for broader use by the scientific community (eg, ClinEpiDB).

The findings of the study will be shared with the community via the local opinion leaders. The project is followed by a community meeting to present some of the preliminary findings. Disease occurrence data will be shared with the MoH. The findings will also be shared in peer-reviewed publications irrespective of final results.

**Author affiliations**
[1]Department of Infection Biology, London School of Hygiene & Tropical Medicine, London, UK
[2]Groupe de Recherche Action en Santé, Ouagadougou, Burkina Faso
[3]Department of Medical Microbiology, Radboud University Nijmegen, Nijmegen, The Netherlands
[4]Pharmacy, Radboudumc, Nijmegen, The Netherlands

**Acknowledgements** The authors thank the Ministry of Health and National Malaria Control Program from Burkina Faso and staff at the study sites for support,

as well as the research nurses, laboratory and administrative staff involved in the study.

**Contributors** ABT, TB and CD developed and conceived the initial design of the study, secured funding and contributed significantly to writing this protocol. MM, AB, MG, SSS, JB, AO, DH and JBY contributed to study design and protocol development. AB, JBY and ABT coordinated the implementation of the study. AB, JBY, SSS, AO, IS, AZO, ES, INO, SBS and DH ran all field operations and provided extensive information on local context and logistics. IS and LG provided drug resistance expertise. DH and JS provided expertise on qualitative methods. IB provided spatial analysis expertise. JR, JB and SS provided statistical expertise. SK and RtH provided pharmacological expertise. MM, MG, TB, CD and ABT wrote the first draft of the manuscript. All authors reviewed and approved the final draft of the manuscript.

**Funding** This work is supported by Bill & Melinda Gates Foundation grant (INV-053846).

**Disclaimer** The funders, Bill & Melinda Gates Foundation, were involved in study design but had no role in data collection, analysis, interpretation or reporting.

**Competing interests** None declared.

**Patient and public involvement** Patients and/or the public were involved in the design, or conduct, or reporting, or dissemination plans of this research. Refer to the Methods section for further details.

**Patient consent for publication** Not applicable.

**Provenance and peer review** Not commissioned; externally peer reviewed.

**ORCID iD**
Marta Moreno http://orcid.org/0000-0002-9091-5612

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
