## [Reviewer comments · BMJ Open]

ARTICLE DETAILS

TITLE (PROVISIONAL)	Understanding and maximizing the community impact of seasonal malaria chemoprevention in Burkina Faso (INDIE-SMC): study protocol for a cluster randomized evaluation trial
AUTHORS	Moreno, Marta; Barry, Aissata; Gmeiner, Markus; Yaro, Jean; Sermé, Samuel; Byrne, Isabel; Ramjith, Jordache; Ouedraogo, Alphonse; Soulama, Issiaka; Grignard, Lynn; Soremekun, Seyi; Koele, S; ter Heine, R; Ouedraogo, Amidou; Sawadogo, Jean; Sanogo, Edith; Ouedraogo, Issa; Hien, Denise; Sirima, Sodiomon Bienvenu; Bradley, John; Bousema, Teun; Drakeley, Chris; Tiono, Alfred B.

VERSION 1 – REVIEW

REVIEWER	Idro, Richard Makerere University College of Health Sciences, Department of Paediatrics and Child Health
REVIEW RETURNED	08-Dec-2023

GENERAL COMMENTS	The study protocol for the: "Understanding and maximizing the community impact of seasonal malaria chemoprevention in Burkina Faso (INDIE-SMC): study protocol for a cluster randomized evaluation trial" makes an interesting reading. Overall, the study aims to better understand factors underlying the lower effectiveness of SMC in real life situations rather than study conditions and what the impact of extending the intervention to school age children <10 years would be on malaria transmission in the region. The abstract is incomplete and largely unstructured. Although a background is given, the objective has to be deduced. Specific aspects of the methods, design, intervention, randomization etc are all not clearly written out. The authors state that a detailed statistical plan is designed but I did not have access to this. The investigators chose to extend the intervention to children <10 years. Was this informed by any specific information on malaria transmission occurring in this age group? From what is presented, one would have expected that they would have taken this up to age 15 years. The sample size and assumptions behind it are unclear. Why was clustering of 3 compounds chosen as a unit? Why not 4 or 5 or other numbers? What may the effect of this be on the primary outcomes? Also, it is unclear what the design effect was in the proposed sample size estimation.
--

	Amodiaquine has some very unpleasant adverse events. Younger children may tolerate it better and may be made to take the medications by the parents but the older children may be less supervised and with the expected unpleasant effects, adherence may be lower. How are the investigators planning to factor such potential scenarios in the trial as it will impact study outcomes and the expected effectiveness?
REVIEWER	Peto, Thomas Mahidol University, Mahidol-Oxford Tropical Medicine Research Unit, Faculty of Tropical Medicine
REVIEW RETURNED	19-Dec-2023
GENERAL COMMENTS	This is a nicely written protocol for an interesting cluster randomized controlled trial that should provide valuable evidence on the effectiveness of extending SMC to children at older ages in a population that is highly vulnerable to malaria. A minor point, but I am not 100% clear why in the section on randomization it states that endpoints are to be evaluated at the level of the individual study participant and not by cluster, please could this sentence be expanded. Overall, this work presents a strong rationale for the need for this study and for its chosen study design. I wish the authors good luck with the implementation.

VERSION 1 – AUTHOR RESPONSE

Response to Reviewer 1

Dr. Richard Idro, Makerere University College of Health Sciences, University of Oxford
Comments to the Author:

The study protocol for the: "Understanding and maximizing the community impact of seasonal malaria chemoprevention in Burkina Faso (INDIE-SMC): study protocol for a cluster randomized evaluation trial" makes an interesting reading. Overall, the study aims to better understand factors underlying the lower effectiveness of SMC in real life situations rather than study conditions and what the impact of extending the intervention to school age children <10 years would be on malaria transmission in the region.

- The abstract is incomplete and largely unstructured. Although a background is given, the objective has to be deduced. Specific aspects of the methods, design, intervention, randomization etc are all not clearly written out.

Response: We thank the reviewer for this comment. We have updated and restructured the abstract aiming at highlighting the 3-arm design, specific intervention per arm, cluster-randomization and main study endpoints accordingly.

- The authors state that a detailed statistical plan is designed but I did not have access to this.

Response: We have included the statistical plan as a supplement.

-The investigators chose to extend the intervention to children <10 years. Was this informed by any specific information on malaria transmission occurring in this age group? From what is presented, one would have expected that they would have taken this up to age 15 years.

Response: There is evidence of seasonal malaria shifting to older children as malaria protection of children under 5 in the Sahel is improving (<https://doi.org/10.1016/j.parepi.2018.02.001>). Moreover, projects in Senegal have shown the effectiveness of SMC in 5- to 10-year-olds (DOI: [10.1371/journal.pmed.1002175](https://doi.org/10.1371/journal.pmed.1002175)).

In 2022, the World Health Organization updated the recommendation on seasonal malaria chemoprevention (SMC) including more flexibility in recognizing age-based risk among children. The National Malaria Control Program (NMCP) from Burkina Faso would like to implement the extension of the program to 10-year-old children in 2024-25 based on evidence of the burden of malaria in children in the country, as shown in previous studies (<http://dx.doi.org/10.2139/ssrn.4585247>). The study protocol presented here was aligned with the plans of the Burkinabe NMCP. Several other countries in the Sahel zone plan to implement the same age extension. The age group up to 15 years has been demonstrated to have a crucial role in malaria transmission, but not necessarily a higher malaria burden) Most importantly, the study we present is aimed to support NMCP decisions by providing evidence of the impact of extended SMC; SMC is currently routinely implemented in children below the age of 5 years with some countries extending it to include children below 10 years of age but none considering an even wider age group. If our study demonstrates success, we do not rule out that future considerations may include further widening the age range.

-The sample size and assumptions behind it are unclear. Why was clustering of 3 compounds chosen as a unit? Why not 4 or 5 or other numbers? What may the effect of this be on the primary outcomes?

Response: The number of compounds per unit is arbitrary and informed by prior experience (see doi: [10.1136/bmjopen-2019-030598](https://doi.org/10.1136/bmjopen-2019-030598)) and feasibility rather than statistical considerations. If individual neighbouring compounds would have been randomized to different arms this is likely to have influenced SMC uptake in the control arm where directly observed treatment (DOT) in nearby compounds would have influenced adherence in the control (standard of care) compounds. In addition to the feasibility of the study, visiting compounds and providing DOT to a large number of children under considerable time-pressure, this was a primary reason to allocate neighbouring compounds to the same arm. Of note, we will not be able to completely prevent a possible impact of the presence of study teams in the area on SMC adherence. This is why we also measure SMC adherence by measuring drug concentrations in children receiving NMCP- implemented SMC outside the study area. We have clarified the reasons for 3 compounds per cluster in the revised manuscript.

-Also, it is unclear what the design effect was in the proposed sample size estimation.

Response: This was taken into account; all analyses include a random effect for cluster. Also, this is clarified in the revised manuscript.

-Amodiaquine has some very unpleasant adverse events. Younger children may tolerate it better and may be made to take the medications by the parents but the older children may be less supervised and with the expected unpleasant effects, adherence may be lower. How are the investigators planning to factor such potential scenarios in the trial as it will impact study outcomes and the expected effectiveness?

Response: In study arm 1 (children under 5), SMC is implemented by the Ministry of Health and will have only the first dose of the treatment given under supervision. In study arms 2 and 3 treatment will be administered with directly observed treatment for the full course of SMC to maximize adherence. We acknowledge that adherence is one of the factors that will impact the effectiveness of the SMC strategy and

also acknowledge that potential side effects of the drugs used for SMC may impact adherence. These are important factors for which the impact is currently unparametrized. Our trial will provide important information on these factors by directly quantifying the difference between arms 1 and 2 (routine practice versus DOT), measuring drug concentrations, and by a qualitative (social science) study component.

This qualitative component aims at capturing adherence to and perceptions about SMC strategy. This includes SMC coverage, health workers and caregivers' compliance and perceptions and will therefore also provide data on reasons for non-compliance including side effects to the SMC drugs. The concerns the reviewer mentions are therefore genuine but will not affect the validity of the comparisons, they are accounted for an in fact a reason to perform this study and better understand why the impact of routine SMC implementation is typically much smaller than expected based on controlled study conditions.

Response to Reviewer 2

Dr. Thomas Peto, Mahidol University

Comments to the Author:

This is a nicely written protocol for an interesting cluster randomized controlled trial that should provide valuable evidence on the effectiveness of extending SMC to children at older ages in a population that is highly vulnerable to malaria.

A minor point, but I am not 100% clear why in the section on randomization it states that endpoints are to be evaluated at the level of the individual study participant and not by cluster, please could this sentence be expanded.

Response: We thank the reviewer for this observation. The endpoint is if people are infected with malaria, and that has to be measured at an individual level. Parasite prevalence will be compared between arms by logistic regression and models will be account for clustering. We have now removed the sentence from the manuscript to avoid confusion. Moreover, a statistical analysis plan (SAP) of the study has been included as supplementary material for more detailed information We have now removed the sentence from the manuscript to avoid confusion.